# Statin Use in Cancer Patients with Acute Myocardial Infarction and Its Impact on Long-Term Mortality

**DOI:** 10.3390/ph15080919

**Published:** 2022-07-25

**Authors:** Konrad Stepien, Karol Nowak, Natalia Kachnic, Grzegorz Horosin, Piotr Walczak, Aleksandra Karcinska, Tomasz Schwarz, Mariusz Wojtas, Magdalena Zalewska, Maksymilian Pastuszak, Bogdan Wegrzyn, Jadwiga Nessler, Jarosław Zalewski

**Affiliations:** 1Department of Coronary Artery Disease and Heart Failure, Jagiellonian University Medical College, 31-202 Krakow, Poland; k.nowak.uj@gmail.com (K.N.); nataliakachnic3103@gmail.com (N.K.); grzegorz.horosin@gmail.com (G.H.); piotr01.walczak@gmail.com (P.W.); aleksandra.karcinska@student.uj.edu.pl (A.K.); jnessler@interia.pl (J.N.); jzalewski@szpitaljp2.krakow.pl (J.Z.); 2Cardiocare, 31-159 Krakow, Poland; rzschwar@cyf-kr.edu.pl (T.S.); m.wojtas@cardiocare.com.pl (M.W.); 3“Club 30”, Polish Cardiac Society, 00-193 Warsaw, Poland; 4Department of Genetics, Animal Breeding and Ethology, University of Agriculture in Krakow, 31-059 Krakow, Poland; 5Department of Emergency Medicine, Faculty of Health Sciences, Jagiellonian University Medical College, 31-126 Krakow, Poland; magdalena.zalewska@uj.edu.pl; 6Department of Pulmonology, District Hospital, 34-600 Limanowa, Poland; maxpast@op.pl; 7Department of Social Work, University of Applied Sciences in Tarnow, 33-100 Tarnow, Poland; bogdnwegrzyn@tlen.pl

**Keywords:** myocardial infarction, statins, cancer, cardio-oncology, prognosis

## Abstract

Statin use and its impact on long-term clinical outcomes in active cancer patients following acute myocardial infarction (MI) remains insufficiently elucidated. Of the 1011 consecutive acute MI patients treated invasively between 2012 and 2017, cancer was identified in 134 (13.3%) subjects. All patients were observed within a median follow-up of 69.2 (37.8–79.9) months. On discharge, statins were prescribed less frequently in MI patients with cancer as compared to the non-cancer MI population (79.9% vs. 91.4%, *p* < 0.001). The most common statin in both groups was atorvastatin. The long-term mortality was higher in MI patients not treated vs. those treated with statins, both in non-cancer (29.5%/year vs. 6.7%/year, *p* < 0.001) and cancer groups (53.9%/year vs. 24.9%/year, *p* < 0.05), respectively. Patient’s age (hazard ratio (HR) 1.04, 95% confidence interval (CI) 1.03–1.05, *p* < 0.001, per year), an active cancer (HR 2.42, 95% CI 1.89–3.11, *p* < 0.001), hemoglobin level (HR 1.14, 95% CI 1.09–1.20, *p* < 0.001, per 1 g/dL decrease), and no statin on discharge (HR 2.13, 95% CI 1.61–2.78, *p* < 0.001) independently increased long-term mortality. In MI patients, simultaneous diagnosis of an active cancer was associated with less frequently prescribed statins on discharge. Irrespective of cancer diagnosis, no statin use was found as an independent predictor of increased long-term mortality.

## 1. Introduction

The current guidelines of the European Society of Cardiology (ESC) on myocardial infarction (MI) suggest lipid-lowering drugs as a key class in secondary prevention. As has been indicated, statins are recommended for all MI patients and should be included in the treatment regimen as soon as possible [1,2]. Moreover, in the subsequent guidelines, there has been a clear trend toward lower target values of lipid parameters. The strong recommendation for the use of statins in patients after MI results from numerous well-controlled studies, in which their beneficial effect on long-term prognosis remains indisputable [3].

Patients with active cancer constitute a special group of MI patients. Traditionally, the diagnosis of potentially lethal neoplastic disease leads to the reduction or even cessation of treatment of chronic comorbidities, including dyslipidemia. However, over the past few decades, due to the significant progress in anti-cancer treatment, more and more patients become long-term survivors. Currently, the approach in this population associated with revascularization and concomitant pharmacotherapy is usually individualized. It is important since the neoplastic disease patients as well as cancer survivors have increased cardiovascular risk of heart failure, arrhythmias, or coronary artery disease [4]. Significant differences may be associated with the oncological characteristics, the type of anti-cancer treatment, as well as the life expectancy. Nevertheless, the current ESC guidelines do not address this group of patients [1,2]. Importantly, patients with active neoplastic disease are most often excluded from randomized controlled trials [3]. The ESC expert document on cancer treatments and cardiovascular toxicity also does not address the issue of optimal lipid-lowering treatment in patients with active cancer and MI [5], despite surprisingly reported lower levels of low-density lipoprotein (LDL) cholesterol in patients with MI and cancer than non-cancer ones (risk ratio 0.87, 95% confidence interval (CI) 0.85–0.89) [6] and the fact that neoplastic treatment changes the lipid profile to pro-atherogenic [7,8,9]. This problem is especially urgent since statins, due to their pleiotropic effects, constitute another potentially useful anticancer pharmacotherapy [10]. Thus far, the data obtained from clinical trials need further evaluation. However, they provide promising results regarding reduced cancer-related mortality in patients taking statins [11]. In a meta-analysis of observational studies analyzing the influence of statins on mortality in the general cancer population, Zhong et al. found a significant reduction of death from any cause (hazard ratio (HR) 0.81, 95% CI 0.72–0.91) as well as decreased cancer-related mortality (HR 0.77, 95% CI 0.66–0.88) in patients treated with statins [12]. However, the available data regarding the impact of statins on MI cancer patients’ survival are still limited.

This study aimed to analyze the statin regimen, including the reasons for withdrawal from statin therapy, in MI patients with active cancer based on real-world data derived from a tertiary cardio-oncology center registry. We also sought to investigate the impact of statin use on long-term mortality.

## 2. Results

### 2.1. Baseline Characteristics

Of 1011 consecutive acute MI patients, cancer was found in 134 (13.3%) subjects, including newly diagnosed cancer in 24 of them (17.9%). Genitourinary, lung, and gastrointestinal were the most frequent cancer subtypes. The remaining 877 (86.7%) acute MI patients without cancer served as controls (Table 1). 

Compared with the non-cancer group, MI patients with cancer were older (*p* = 0.004) and had a lower body mass index (BMI) (*p* = 0.012). Patients with neoplasm had lower hemoglobin (*p* < 0.001) and hematocrit (*p* < 0.001) (Table 2). 

Therefore, the diagnosis of anemia (*p* < 0.001) as well as thrombocytopenia (*p* = 0.03) was more frequent in the latter group (Table 1). Cancer subjects were characterized by higher-sensitivity troponin levels on admission (*p* < 0.001) as well as peak values (*p* = 0.013), without significant differences in respective levels of creatine kinase MB isoenzyme.

In cancer patients, left ventricular ejection fraction (LVEF) was lower than in non-cancer ones (*p* = 0.008). The analysis of coronary angiography of cancer patients revealed more common distal embolization (*p* = 0.004) with a lower prevalence of obstructive coronary lesions (*p* < 0.001) as compared to non-cancer subjects. Both invasive therapeutic strategies including percutaneous coronary intervention or coronary artery bypass surgery were more frequent in non-cancer than in cancer patients (*p* = 0.023) (Table 1).

### 2.2. Lipid Profile, Dyslipidemia, and Use of Statins in Cancer vs. Non-Cancer Patients

Dyslipidemia was identified less frequently in cancer than in non-cancer patients (*p* < 0.001) (Table 1). Patients with cancer had lower plasma levels of total cholesterol (*p* = 0.006), high-density lipoprotein (HDL) cholesterol (*p* < 0.001), and triglycerides (*p* = 013), without significant differences in LDL cholesterol levels (*p* = 0.70) (Table 2). 

As shown in Figure 1, statins were less frequently prescribed on discharge in MI patients with cancer than in non-cancer subjects (79.9% vs. 91.4%, *p* < 0.001).

Their distribution was also different in both groups. However, the most commonly used statin in both groups was atorvastatin (72.9% vs. 75.2%, respectively, *p* = 0.64). The second most frequent statin in the cancer group was simvastatin, while in the non-cancer group, it was rosuvastatin.

The multivariable logistic regression analysis indicated that in the whole study group, a lack of coronary artery stenosis >50% (odds ratio (OR) 4.47, 95% confidence interval (95% CI) 2.13–9.40, *p* < 0.001), active cancer (OR 2.13, 95% CI 1.04–4.35, *p* = 0.038), and anemia (OR 1.89, 95% CI 1.01–3.57, *p* = 0.045) (Table 3) independently decreased the chance of prescription of statins on discharge, while a preserved glomerular filtration rate increased this possibility (OR 0.98, 95% CI 0.96–0.99, *p* = 0.006 per 1 mL/min). 

Hypertension and LDL cholesterol as traditional cardiovascular risk factors were associated with more frequent use of statins, but only in a univariate model. 

In the cancer subgroup, hypertension independently favored statin usage, while in non-cancer patients, a lack of coronary artery stenosis >50% and anemia were associated with a lower chance of statin prescription. 

### 2.3. Insights into Oncological Subgroups

Cancer patients with vs. without lymph node metastases had lower plasma HDL cholesterol (1.0 [0.9; 1.1] vs. 1.2 [1.0; 1.4] mmol/L, *p* = 0.047), without differences in total cholesterol (*p* = 0.44), LDL cholesterol (*p* = 0.87), and triglycerides (*p* = 0.76). No differences in lipid profile were found between cancer patients with vs. without distant metastases. Patients with prior chemotherapy had lower levels of hemoglobin (12.9 [12.2; 14.1] vs. 11.3 [10.5; 12.5] g/dL, *p* < 0.001), while patients with a history of radiotherapy had more frequent strokes (25% vs. 6.8%, *p* = 0.017). The frequency of statin use, and their type in patients with vs. without lymph node metastases, distant metastases, and history of radiotherapy or chemotherapy, were comparable.

Cancer patients who were treated with statins had a higher glomerular filtration rate (70 [60; 82] vs. 57 [45; 73] mL/min, *p* = 0.042) and higher hemoglobin levels (12.9 [11.8; 14.0] vs. 11.8 [10.9; 13.0] g/dL, *p* = 0.045), but without significant differences in lipid profile as compared to those without statin therapy. They also more frequently received P2Y12 inhibitors (*p* = 0.030) or angiotensin-converting enzyme inhibitor/angiotensin receptor blocker (*p* = 0.011) than patients without statins.

### 2.4. Long-Term Mortality and Its Determinants

As expected, the higher long-term mortality was demonstrated in the cancer MI group than in non-cancer patients (71.6 vs. 30.9%, *p* < 0.001). There was also a different distribution of causes of death in these two groups (*p* < 0.001). The most frequent cause of death in cancer MI patients was cancer (52 patients, 54% of deaths) and the second was cardiovascular diseases (30, 31.3%), including coronary artery disease, heart failure, stroke, or atherosclerosis. In contrast, among non-cancer MI patients, cardiovascular deaths were the most common (157, 57.9%), the second cause was other diseases (66, 24.4%), and cancer was the third (48, 17.7%). 

The mortality rates were also significantly higher in MI patients not treated with statins, both in the non-cancer population (29.5%/year vs. 6.7%/year, *p* < 0.001) as well as in the cancer group (53.9%/year vs. 24.9%/year, *p* < 0.05) when compared to respective groups treated with statins (Figure 2). 

The Cox regression analysis showed that a higher patient’s age (hazard ratio (HR) 1.04 95% CI 1.03–1.05, *p* < 0.001, per 1 year), active cancer (HR 2.42, 95% CI 1.89–3.11, *p* < 0.001), the presence of at least one coronary artery stenosis > 50% (HR 1.86, 95% CI 1.23–2.80, *p* = 0.003), and a higher plasma creatinine level (HR 1.002, 95% CI 1.001–1.003, *p* = 0.011, per 1 µmol/L) independently increased long-term mortality in the whole sample (Table 4).

In contrast, better left ventricular ejection fraction (HR 0.97, 95% CI 0.96–0.98, *p* < 0.001, per 1%), statin prescription on discharge (HR 0.47, 95% CI 0.36–0.62, *p* < 0.001), hypertension (HR 0.50, 95% CI 0.37–0.65, *p* < 0.001), and higher hemoglobin concentration (HR 0.88, 95% CI 0.83–0.92, *p* < 0.001, per 1 g/dL) were associated with improved long-term survival. 

In the cancer subgroup, increased long-term mortality was influenced by the patients’ age and the presence of at least one coronary artery stenosis >50%, while a protective effect was associated with hypertension and statin prescription on discharge (Table 4).

### 2.5. Statins in MI with Non-Obstructive Coronary Arteries (MINOCA) Subgroup

Statins were used by 54 (75.0%) of the MINOCA subjects (Table 5).

MINOCA patients treated with statins had higher prevalence of hypertension (*p* = 0.001), dyslipidemia (*p* < 0.001), lower Killip class on admission (*p* = 0.005), and a higher LVEF (*p* = 0.019) than MINOCA patients not treated with statin. At the time of decision about statin therapy, the LDL cholesterol level was higher in the MINOCA population with subsequently prescribed statins (*p* = 0.008). Moreover, P2Y12 inhibitors were more often prescribed in the statin group (*p* = 0.047). 

Long-term mortality was significantly higher in MINOCA patients not treated with statins (17.7%/year vs. 6.6%/year, *p* = 0.009) compared to those treated with statins (Figure 3). The most frequent cause of death in the whole MINOCA subgroup was cardiovascular diseases (14 patients, 48.3%). Deaths associated with cancer were found in 9 (31.3%) patients, while the remaining 6 (20.7%) subjects died due to other disorders. There were no significant differences in the distribution of causes of death between statin and non-statin MINOCA patients within the first 12 months after MI (*p* = 0.63) as well as later (*p* = 0.24). The most common cause of death in the first period following MI in statin vs. non-statin MINOCA groups was cardiovascular disorders (57.1% vs. 50.0%, respectively), followed by cancer-related death (28.6% vs. 50.0%, respectively). In the statin group, in one case the cause of death remained unknown. After the first year elapsed since index MI, in the statin group, cancer was the most frequent cause of death (4, 44.4%), followed by cardiovascular diseases (3, 33.3%), while in patients without statins, cardiovascular diseases were the most common (6, 66.6%), followed by 1 (11.1%) death associated with cancer. 

## 3. Discussion

This study’s findings demonstrate that active cancer patients with MI are characterized by a lower statin prescription rate as compared to non-oncological patients. Although significant differences in the baseline lipid profile were observed between cancer and non-cancer groups, statin prescription was not associated with the LDL cholesterol level. The decision of no statin initiation was based on clinical characteristics, including the presence of active cancer, anemia, a reduced renal function, and a lack of significant coronary lesions. In cancer patients, a history of hypertension favored more frequent statin prescription. Irrespective of the occurrence of neoplastic disease, patients without prescribed statins on discharge were characterized by higher long-term mortality, which was confirmed in multivariable analysis. The analyses limited to the MINOCA population indicate that statin users in this subgroup of patients had better long-term survival without the statistically significant differences in causes of patients’ deaths.

In recent years, the interaction between statins and cancer has been thoroughly investigated. As shown in some studies, the use of statins was associated with an increased risk of breast or gastrointestinal cancer [13,14]. These disturbing signals, as was mentioned in the Section 1, have not been confirmed in the subsequent studies nor in the developed meta-analyses [15]. According to the current guidelines, the carcinogenic effect has also not been confirmed with the achievement of extremely low lipid parameters [3]. In contrast, statins are considered cardioprotective drugs due to their pleiotropic mechanisms of action, and therefore they are included as one of the strategies to reduce chemotherapy-induced cardiotoxicity in the current ESC expert document on cancer treatment [5]. It is worth paying attention to the interesting results from preclinical studies qualifying statins as potential chemotherapeutic agents [16] in different cancer subtypes [17]. In the proposed anti-cancer mechanism, statins inhibit the cell cycle proteins such as cyclins, the TNF-α synthesis, and the metastatic process by the downregulation of metalloproteinases [10,18,19].

As mentioned above, there are limited data on the clinical importance of statins’ use in patients with MI and active cancer. Patients with cancer are characterized by a three-fold higher risk of MI, which depends on the type of neoplasm and its oncological features [20]. The suggested mechanism of this deleterious relationship is an increased prothrombotic tendency observed in these patients [21,22]. Most likely, however, the malignancy-associated dyslipidemia is also an important reason [23]. The results of a retrospective study of patients with cancer and acute MI from the leading cardio-oncological center showed that neither statin therapy nor catheter-based revascularization had a significant impact on mortality, contrary to aspirin and beta-blockers that decreased the risk of death. Importantly, hyperlipidemia was a protective factor for one-year survival [24]. Koo et al. demonstrated that in patients with prior cancer and subsequent MI, lower values of LDL cholesterol, total cholesterol, and triglycerides have been observed [6]. Our results showed lower total cholesterol and HDL cholesterol without significant differences in LDL cholesterol in active cancer patients as compared to non-cancer subjects. For many years, the hypolipidemia observed in cancer patients has been an extremely unfavorable prognostic factor [25]. So far, there are several potential mechanisms of this phenomenon, such as the direct lipid-lowering effect of tumor cells, cancer-related malfunction of the lipid metabolism, as well as the effect of antioxidant vitamins or chemotherapeutic agents used in the cancer treatment [23]. It may also reflect cachexia common in this patient population [23]. The relationship between serum lipid levels and cancer progression remains unclear as available studies provide conflicting information.

The important issue is the reason for no statin prescription, especially in MI survivors qualified as having the highest cardiovascular risk. According to our study, only hypertension was an independent factor favoring statin use in cancer patients, however univariate comparisons suggest that patients who were treated with statins could be in better health condition than the non-statin population. It is likely associated with the proper time of therapy cessation during neoplastic disease progression and worsening of health status. Kutner et al. have found that in patients with limited oncological prognosis with a survival time of 1 month to 1 year, cessation of statin therapy is safe in terms of 60-day mortality, cardiovascular events, and performance status. It can be associated with improved quality of life and the use of fewer non-statin medications [26]. Moreover, Frisk et al. suggested that earlier statin discontinuation in women with advanced cancer did not affect cardiovascular mortality, and thus it can also be applied to men with advanced cancer [27]. A hypothesis that the use of statins can be considered as an indicator of oncological well-being requires prospective studies regarding the statin regimen, the proper time of its potential discontinuation, and its clinical relevance in high-risk cancer patients. 

MINOCA is a complex disease entity, and the optimal pharmacological treatment regimen has not been finally developed [28]. Currently, due to the lack of visible atherosclerosis in coronary arteries, statins are prescribed less frequently in the MINOCA group than in MI patients with obstructive coronary artery disease [29]. We have shown that mortality was lower among MINOCA patients taking statins compared to those without statin therapy. The available data remain inconsistent [30]. To date, there are also no published randomized clinical trials evaluating the use of statins in this population. However, in a recent meta-analysis of six observational studies, Masson et al. clearly demonstrated the positive effect of statins on the MINOCA patients’ prognosis [31].

To date, the clinical relevance of statin use in MI patients with an active cancer is limited and often inconsistent. Our findings indicate that statins are less frequently prescribed in this group of patients at discharge. Using the real-life character of our registry, we have identified several factors potentially important in the decision-making process associated with omitting statins in chronic treatment. When analyzing the long-term mortality, no statin use was an unambiguous factor of an unfavorable prognosis even in patients without cancer. A similar association was found in patients with MINOCA, which confirms the important position of statins in the current guidelines in this group of patients.

Our study has several limitations. First, we focused only on the treatment at the time of discharge from the hospital. We did not analyze the patients’ compliance as well as follow-up pharmacotherapy modifications that could have been updated by telephone contact. Second, apart from all-cause mortality, we did not analyze the other clinical endpoints, such as cardiovascular mortality, subsequent MI, heart failure decompensation, or revascularization. In the cancer population, no information on recurrent surgery, chemotherapy, or radiotherapy has been collected. Third, due to the limitations of our registry, we did not present the data regarding liver function, smoking, and mental state [32]. Finally, we did not perform the recommended cardiac magnetic resonance and intracoronary imaging in the MINOCA patients due to the undetermined and debated diagnostic significance at the time of enrolling patients in the registry.

## 4. Materials and Methods

In our tertiary cardio-oncology center, 1011 patients were hospitalized between 2012 and 2017 with the diagnosis of MI based on clinical symptoms, electrocardiographic findings, and the evolution of myocardial necrotic biomarkers according to the universal criteria of MI. On admission, coronary angiography was performed in all patients to assess the presence of obstructive lesions in coronary arteries. Patients with ST-segment elevation of at least 1 mm in at least two contiguous leads were classified as ST-segment elevation MI (STEMI), whereas patients without ST-segment elevation on admission were diagnosed as non-ST-segment elevation MI (NSTEMI) [33]. 

The active malignancy was identified based on data from the previous medical history, or de novo based on detailed clinical examination during the current hospitalization. The active disease was defined as cancer diagnosed within the past 6 months, receiving antimitotic treatment during the last 6 months, recurrent, metastatic, regionally advanced, or inoperable [34].

### 4.1. Patients’ Characteristics

The data including patients’ demographics, anthropometric parameters, cardiovascular risk factors, the history of cardiovascular diseases, comorbidities, and concomitant medications were collected. Additionally, a detailed oncological history was gathered. Anemia was defined as hemoglobin level < 13 g/dL in men or <12 g/dL in women [35], whereas thrombocytopenia as a platelets level lower than 100 × 10^3^/μL [36]. The pre-end-stage renal disease and end-stage renal disease were established when creatinine clearance calculated using the Cockcroft–Gault formula was lower than 30 mL/min. Cardiac necrotic biomarkers, including isoenzyme MB of creatine kinase (IU/L, upper limit of normal of 24 IU/L) and the highly sensitive cardiac troponin T (ng/mL, upper limit of normal: 0.014 ng/mL), were measured on admission and at least once within the first 24 h. According to the latest guidelines, during the index hospitalization dyslipidemia was defined as elevated levels of total cholesterol or low-density lipoproteins or current hypolipidemic treatment [37]. The length of index hospitalization was calculated based on hospital records, whereas data concerning long-term all-cause mortality were derived from the Polish National Health Registry. The study protocol complied with the Declaration of Helsinki and was approved by the Jagiellonian University Medical College Ethics Committee (Consent No. 1072.6120.59.2018). All included patients gave informed consent.

### 4.2. Angiography

All coronary angiograms were analyzed using two contralateral projections for each artery at baseline and after angioplasty if applicable, by a cardiologist unaware of the clinical data. All coronary segments were evaluated for stenosis, distal coronary embolization, and epicardial thrombus based on visual inspection [38]. In case of suspicious borderline obstructive lesions between 40% and 70%, quantitative coronary angiography (QCA Quantcor, Siemens, Germany) was performed for detailed assessment. Lesions narrowing a coronary artery by less than 50% were defined as non-obstructive according to the current guidelines [39] and patients were presumed as MINOCA [40,41]. 

### 4.3. Statistical Analysis

Statistical analysis was performed with the SPSS Statistics software (Version 25.0.0.2, IBM Corp., Armonk, N.Y., USA). Continuous variables were expressed as medians (interquartile range) and categorical variables as numbers (percentage). Continuous variables were first checked for normal distribution using the Shapiro–Wilk test. Afterwards, differences in the groups among continuous variables were compared by the Student’s t-test or Mann–Whitney U test if the distribution was normal or different than normal, respectively. Categorical variables were analyzed with the chi-square test or Fisher’s exact test. Kaplan–Meier curves for overall mortality were constructed to estimate the survival rates, and a log-rank test with a Bonferroni-corrected threshold was performed to assess the differences in survival between the studied groups. Finally, all independent variables with the potential to confound both the exposure and the outcome were included in the logistic regression analysis to find independent predictors of no statin use or in the Cox proportional hazard regression model to determine independent predictors of long-term all-cause mortality. A two-tailed *p*-value of less than 0.05 was considered statistically significant.

## 5. Conclusions

Our findings demonstrated that in myocardial infarction patients, simultaneous diagnosis of an active cancer was associated with less frequently prescribed statins on discharge. Irrespective of cancer diagnosis, no statin use was found as an independent predictor of increased long-term mortality. Simultaneously, statin use was also associated with favorable prognosis in MINOCA patients.

## Figures and Tables

**Figure 1 pharmaceuticals-15-00919-f001:**
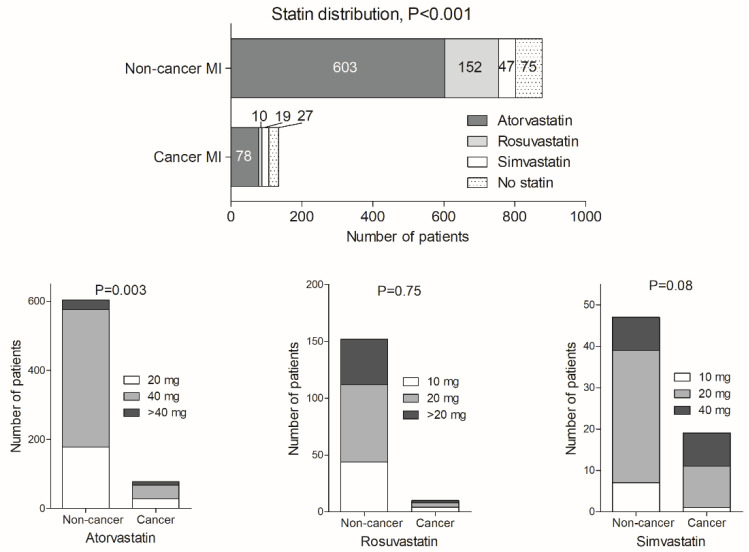
Statins’ distribution and their doses in myocardial infarction patients with and without cancer. Abbreviations: MI: myocardial infarction, *p*-value for differences between cancer and non-cancer patients.

**Figure 2 pharmaceuticals-15-00919-f002:**
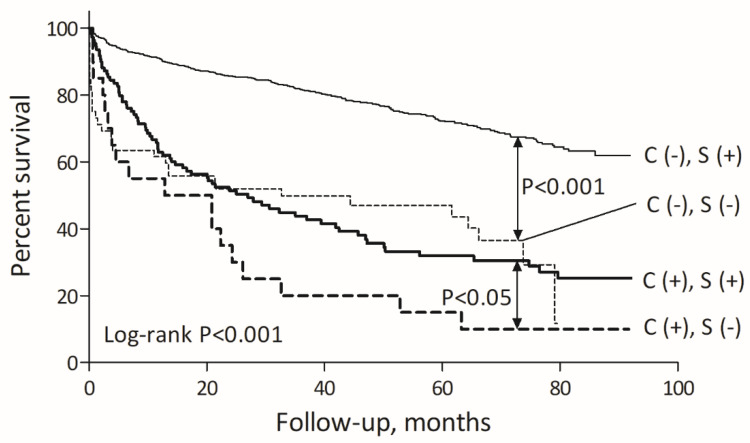
The impact of active cancer and statin usage on long-term mortality. Abbreviations: C: cancer, S: statin.

**Figure 3 pharmaceuticals-15-00919-f003:**
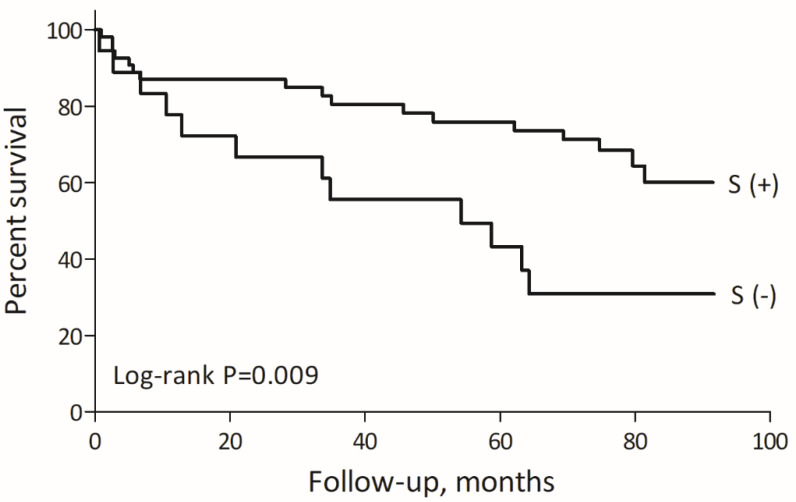
The long-term mortality in MINOCA patients treated with vs. without statins. Abbreviations: S: statin.

**Table 1 pharmaceuticals-15-00919-t001:** Clinical characteristics of the studied patients.

	Cancer MI	Non-Cancer MI	*p*-Value
	*n* = 134	*n* = 877	
Male gender	96 (71.6)	618 (70.5)	0.78
Age, years	73 (66; 79)	68 (60; 78)	0.004
Body mass index, kg/m^2^	27.1 (23.5; 30.1)	27.7 (24.9; 30.9)	0.012
Diabetes mellitus	47 (35.1)	331 (37.9)	0.53
Hypertension	112 (83.6)	755 (86.4)	0.38
Dyslipidemia	85 (63.4)	742 (84.9)	<0.001
Pre-ESRD or ESRD	7 (5.22)	22 (2.51)	0.11
Anemia	61 (45.5)	180 (20.5)	<0.001
Thrombocytopenia	5 (3.7)	11 (1.3)	0.030
Prior myocardial infarction	42 (31.4)	247 (28.3)	0.46
Prior stroke	12 (9.0)	59 (6.8)	0.35
Killip class on admission:			0.10
I/II	117 (87.3)	804 (91.7)	
III/IV	17 (12.7)	73 (8.3)	
Clinical presentation:			0.84
NSTEMI	89 (66.4)	575 (65.6)	
STEMI	45 (33.6)	302 (34.4)	
LVEF, %	45 (37; 55)	50 (40; 55)	0.008
Type of cancer:			
Genitourinary	44 (32.8)	-	
Breast	12 (9.0)	-	
Lung	31 (23.1)	-	
Gastrointestinal	22 (16.4)	-	
Other	25 (18.7)	-	
Metastatic disease:			
Lymph nodes	16 (11.9)	-	
Distant	28 (20.9)	-	
Prior oncological treatment:			
Surgery	30 (22.4)	-	
Surgery with curative intent	4 (3.0)	-	
Radiotherapy	16 (11.9)	-	
Chemotherapy	32 (23.9)	-	
Platinum compounds	11 (8.2)	-	
Taxanes	4 (3.0)	-	
Fluoropyrimidines	10 (7.5)	-	
Anthracyclines	3 (2.2)	-	
Other	4 (3.0)	-	
Hormonotherapy	19 (14.2)	-	
Newly diagnosed cancer	24 (17.9)	-	
Coronary angiography:			
≥50% stenosis	113 (84.3)	826 (94.2)	<0.001
Epicardial thrombus	14 (10.4)	117 (13.3)	0.35
Distal embolization	9 (6.7)	20 (2.3)	0.004
Treatment strategy:			0.074
Percutaneous coronary intervention	101 (75.4)	724 (82.6)	
Coronary artery bypass graft surgery	3 (2.2)	24 (2.7)	
Optimal medical treatment	30 (22.4)	129 (14.7)	
Pharmacotherapy:			
Aspirin	127 (94.8)	854 (97.3)	0.17
P2Y12 inhibitor	115 (85.8)	812 (92.6)	0.008
Proton pump inhibitor	92 (68.7)	652 (75.0)	0.11
ACEI/ARB	120 (89.6)	763 (87.0)	0.41
β-blocker	117 (87.3)	780 (89.7)	0.39
Statin	107 (80.5)	801 (92.1)	<0.001

Abbreviations: data are shown as median (interquartile range) or number (percentage), ACEI: angiotensin-converting enzyme inhibitor, ARB: angiotensin receptor blocker, ESRD: end-stage renal disease, LVEF: left ventricular ejection fraction, MI: myocardial infarction, NSTEMI: non-ST-segment elevation myocardial infarction, STEMI: ST-segment elevation myocardial infarction.

**Table 2 pharmaceuticals-15-00919-t002:** Laboratory characteristics.

	Cancer MI	Non-Cancer MI	*p*-Value
	*n* = 134	*n* = 877	
Hemoglobin, g/dL	12.8 (11.2; 14.0)	13.8 (12.8; 15.0)	<0.001
Hematocrit, %	38.5 (33.9; 41.6)	41.2 (38.3; 44.5)	<0.001
White blood cells, ×10^3^/µL	9.7 (7.4; 12.9)	9.3 (7.4; 11.9)	0.34
Platelet count, ×10^3^/µL	237.5 (181.5; 290.5)	221.0 (184.0; 270.0)	0.26
Creatinine, µmol/L	92.5 (77.0; 114.5)	88.0 (76.0; 104.0)	0.11
Glomerular filtration rate, ml/min	65.5 (48; 85)	71 (57; 86)	0.09
Glucose, mmol/L	7.5 (5.7; 8.9)	6.9 (5.8; 9.1)	0.45
Troponin, ng/mL	0.19 (0.05; 1.07)	0.11 (0.03; 0.42)	<0.001
Troponin peak, ng/mL	0.61 (0.15; 6.27)	0.45 (0.14; 1.91)	0.013
Creatine kinase MB isoenzyme, IU/L	24 (15; 51)	22 (15; 42)	0.57
Creatine kinase MB isoenzyme peak, IU/L	41 (22; 119)	36 (19; 98)	0.44
Total cholesterol, mmol/L	4.1 (3.4; 4.8)	4.4 (3.6; 5.3)	0.006
LDL, mmol/L	2.5 (1.9; 3.1)	2.6 (1.7; 3.4)	0.70
HDL, mmol/L	1.1 (0.9; 1.4)	1.2 (1.0; 1.6)	<0.001
Triglycerides, mmol/L	1.1 (0.9; 1.5)	1.3 (0.9; 1.7)	0.013

Abbreviations: data are shown as median (interquartile range), HDL: high-density lipoprotein, LDL: low-density lipoprotein, MI: myocardial infarction.

**Table 3 pharmaceuticals-15-00919-t003:** The multivariable logistic regression of no statin use.

Independent Variables	Univariate Model			Multivariate Model		
All patients, Chi^2^ = 40.1, df = 7, *p* < 0.001						
	*p*-value	OR	95% CI for OR	*p*-value	OR	95% CI for OR
Age, per 1 year	0.017	1.03	1.01–1.04	0.956	1.01	0.97–1.03
Cancer, yes vs. no	<0.001	2.63	1.56–4.35	0.038	2.13	1.04–4.35
Lack of coronary stenosis of >50%, yes vs. no	<0.001	3.80	2.12–6.81	<0.001	4.47	2.13–9.40
Hypertension, yes vs. no	<0.001	0.40	0.24–0.65	0.258	0.65	0.31–1.37
Anemia, yes vs. no	<0.001	2.56	1.67–4.00	0.045	1.89	1.01–3.57
Glomerular filtration rate, per 1 mL/min	<0.001	0.97	0.96–0.98	0.006	0.98	0.96–0.99
LDL cholesterol, per 1 mmol/L	0.041	0.78	0.61–0.99	0.210	0.83	0.63–1.11
Cancer group, Chi^2^ = 10.9, df = 3, *p* = 0.012						
	*p*-value	OR	95% CI for OR	*p*-value	OR	95% CI for OR
Lack of coronary stenosis of >50%, yes vs. no	0.059	2.74	0.96–7.78	0.314	1.89	0.55–6.53
Hypertension, yes vs. no	0.021	0.30	0.11–0.83	0.029	0.28	0.09–0.88
Glomerular filtration rate, per 1 mL/min	0.045	1.02	1.00–1.04	0.091	1.02	0.99–1.04
Non-cancer group, Chi^2^ = 26.9, df = 6, *p* < 0.001						
	*p*-value	OR	95% CI for OR	*p*-value	OR	95% CI for OR
Age, per 1 year	0.037	1.02	1.01–1.06	0.993	1.00	0.97–1.03
Lack of coronary stenosis of >50%, yes vs. no	<0.001	3.61	1.76–7.40	<0.001	5.66	2.36–13.57
Hypertension, yes vs. no	0.006	0.44	0.25–0.79	0.945	0.97	0.36–2.58
Anemia, yes vs. no	<0.001	2.79	1.67–4.67	0.025	2.28	1.11–4.68
Glomerular filtration rate, per 1 mL/min	<0.001	0.97	0.95–0.98	0.008	0.98	0.96–0.99
LDL cholesterol, per 1 mmol/L	0.031	0.74	0.56–0.98	0.200	1.23	0.90–1.67

Abbreviations: LDL: low-density lipoprotein.

**Table 4 pharmaceuticals-15-00919-t004:** The Cox proportional hazard regression of all-cause long-term mortality.

Independent Variables	Univariate Model			Multivariate Model		
All patients, Chi^2^ = 393, df = 12, *p* < 0.001						
	*p*-value	HR	95% CI for HR	*p*-value	HR	95% CI for HR
Age, per 1 year	<0.001	1.05	1.04–1.06	<0.001	1.04	1.03–1.05
Male gender, yes vs. no	0.790	0.97	0.78–1.21	0.898	1.02	0.80–1.29
Body mass index, per 1 kg/m^2^	<0.001	0.95	0.93–0.98	0.311	0.99	0.96–1.01
Active cancer, yes vs. no	<0.001	3.34	2.64–4.22	<0.001	2.42	1.89–3.11
Diabetes mellitus, yes vs. no	0.002	1.39	1.13–1.70	0.111	1.20	0.96–1.01
Hypertension, yes vs. no	0.010	0.70	0.54–0.92	<0.001	0.50	0.37–0.65
Coronary stenosis of >50%, yes vs. no	0.786	1.05	0.72–1.54	0.003	1.86	1.23–2.80
Left ventricular ejection fraction, per 1%	<0.001	0.97	0.96–0.98	<0.001	0.97	0.96–0.98
Hemoglobin, per 1 g/dL	<0.001	0.80	0.77–0.83	<0.001	0.88	0.83–0.92
LDL cholesterol, per 1 mmol/L	<0.001	0.77	0.70–0.85	0.068	0.90	0.80–1.01
Creatinine, per 1 µmol/L	<0.001	1.003	1.002–1.004	0.011	1.002	1.001–1.003
Statin use, yes vs. no	<0.001	0.29	0.22–0.37	<0.001	0.47	0.36–0.62
Cancer group, Chi^2^ = 22.4, df = 6, *p* = 0.001						
	*p*-value	HR	95% CI for HR	*p*-value	HR	95% CI for HR
Age, per 1 year	0.048	1.02	1.00–1.05	0.007	1.04	1.01–1.06
Statin use, yes vs. no	0.004	0.50	0.31–0.81	0.034	0.56	0.32–0.96
Hypertension, yes vs. no	0.089	0.64	0.38–1.07	0.018	0.50	0.28–0.89
Hemoglobin, per 1 g/dL	0.016	0.90	0.82–0.98	0.075	0.91	0.83–1.01
Coronary stenosis of >50%, yes vs. no	0.228	1.42	0.80–2.51	0.037	1.92	1.04–3.53
LDL cholesterol, per 1 mmol/L	0.090	0.78	0.59–1.04	0.677	0.94	0.68–1.28

Abbreviations: LDL: low-density lipoprotein.

**Table 5 pharmaceuticals-15-00919-t005:** Characteristics of MINOCA patients with or without prescribed statins.

	Statin MINOCA	Non-Statin MINOCA	*p*-Value
	*n* = 54	*n* = 18	
Male gender	27 (50.0)	10 (55.6)	0.68
Age, years	72.5 (66; 79)	72 (54; 78)	0.17
Body mass index, kg/m^2^	28.1 (25.1; 31.6)	26.1 (23.0; 29.9)	0.08
Diabetes mellitus	18 (33.3)	2 (11.1)	0.06
Hypertension	46 (85.2)	8 (44.4)	0.001
Dyslipidemia	51 (94.4)	8 (44.4)	<0.001
Pre-ESRD or ESRD	3 (5.6)	0 (0.0)	0.42
Active cancer	14 (25.9)	7 (38.9)	0.45
Killip class on admission:			0.005
I/II	53 (98.1)	13 (72.2)	
III/IV	1 (1.9)	5 (27.8)	
Clinical presentation:			0.27
NSTEMI	47 (87.0)	13 (72.2)	
STEMI	7 (13.0)	5 (27.8)	
LVEF, %	55 (45; 60)	40 (30; 55)	0.019
Lipid profile:			
Total cholesterol, mmol/L	4.4 (3.5; 5.4)	4.0 (3.1; 4.2)	0.06
LDL, mmol/L	2.7 (1.9; 3.8)	1.9 (1.1; 2.6)	0.008
HDL, mmol/L	1.1 (1.0; 1.5)	1.3 (1.1; 2.0)	0.52
Triglycerides, mmol/L	1.3 (0.9; 1.6)	1.1 (0.8; 1.5)	0.69
Pharmacotherapy:			
Aspirin	28 (51.9)	10 (55.6)	0.50
P2Y12 inhibitor	26 (48.2)	4 (22.2)	0.047
Proton pump inhibitor	36 (66.7)	9 (50.0)	0.33

Abbreviations: data are shown as median (interquartile range) or number (percentage), ACEI: angiotensin-converting enzyme inhibitor, ARB: angiotensin receptor blocker, ESRD: end-stage renal disease, HDL: high-density lipoprotein, LDL: low-density lipoprotein, LVEF: left ventricular ejection fraction, MINOCA: myocardial infarction with non-obstructive coronary arteries, NSTEMI: non-ST-segment elevation myocardial infarction, STEMI: ST-segment elevation myocardial infarction.

## Data Availability

Data is contained within the article.

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
