# Peer review of "Statin Use in Cancer Patients with Acute Myocardial Infarction and Its Impact on Long-Term Mortality"

_pharmaceuticals, 2022, doi:10.3390/ph15080919_

Round 1

Reviewer 1 Report

The manuscript titled “Statins use in cancer patients with acute myocardial infarction and its impact on long-term mortality” has been well summarized, however, there are a few minor issues with the manuscript as mentioned below

·         In the abstract if the author is including abbreviation terms, then it would be helpful to mention at the bottom of the abstract some keywords that would help to identify the full form for the abbreviation.

·         In the Discussion section the authors have mentioned in the last paragraph a very short summary of the limitations to their study. These limitations should have been a little more elaborated and what alternatives could be done to take care of the limitations should be clearly mentioned. The authors could include the novelty of their study within a separate “significance” section.

·         MINOCA has been mentioned as a complex disease but when the authors first state about it they should mention the full form for the abbreviation.

Author Response

1. In the abstract if the author is including abbreviation terms, then it would be helpful to mention at the bottom of the abstract some keywords that would help to identify the full form for the abbreviation.

As requested, all abbreviations used in the Abstract has been explained at the first mention.

2. In the Discussion section the authors have mentioned in the last paragraph a very short summary of the limitations to their study. These limitations should have been a little more elaborated and what alternatives could be done to take care of the limitations should be clearly mentioned. The authors could include the novelty of their study within a separate “significance” section.

The Limitations section has been elaborated. As you suggested, we also added kind of “Significance” paragraph at the end of Discussion just before limitations.

3. MINOCA has been mentioned as a complex disease but when the authors first state about it they should mention the full form for the abbreviation.

Done.

Reviewer 2 Report

The manuscript of Stepien et al. describes the effect of stain use in cancer patients after an acute myocardial infarction (MI) with an average followup time of 69 months. This is an important issue to ascertain whether statins are beneficial in these cancer patients who have had an acute MI. Similar to previous publish papers, the authors find a significant benefit to these patients taking statins by a decreased all-cause mortality. I find the paper well written and clear. Statistics are sound. I find the conclusions well supported by the data. I have only a few points that need to be addressed.  
  1. Figure 3 is very instructive. The deviation of the trend of statin use or not occurs at about 8 months. So, I believe the authors have a very important piece of information about how the people died. What happened after 8 months? The authors know the cause of death and should share this information to demonstrate whether there is an increase in cancer recurrence or side effect from the cancer (which statin inhibits), or whether there is an increase in heart disease related deaths. This is a very important point and will make a big impact in this paper.
  2. Figure 2: the lines must be clearly identified. I am not sure which line represents C (-) S (-). If it is the dotted line, then statin use is almost as big a determinant of survival as active cancer! The authors need to address this point.
  3. Reference 12 uses a meta-analysis of many previous studies to come to a similar conclusion. The authors need to highlight what is different about their cohort or study, or what this study adds in addition to the previous studies. Do these results apply specifically to the population in this study?

Author Response

1. Figure 3 is very instructive. The deviation of the trend of statin use or not occurs at about 8 months. So, I believe the authors have a very important piece of information about how the people died. What happened after 8 months? The authors know the cause of death and should share this information to demonstrate whether there is an increase in cancer recurrence or side effect from the cancer (which statin inhibits), or whether there is an increase in heart disease related deaths. This is a very important point and will make a big impact in this paper.

Thank you for this important comment. The suggested analysis was included in the paragraph devoted to MINOCA. In our study we did not find significant differences in terms of particular mortality causes (cancer, cardiovascular or others) both within the first 12 months after index MI or later. Perhaps the lack of such a difference results from the small sample size of the MINOCA group.

2. Figure 2: the lines must be clearly identified. I am not sure which line represents C (-) S (-). If it is the dotted line, then statin use is almost as big a determinant of survival as active cancer! The authors need to address this point.

The Figure 2 has been corrected. As we shown in multivariable model, the lack of statin use in patients with MI diagnosis which is a strong cardiovascular indication for their use, is rather an indicator of poor clinical patient’s condition, fragile syndrome or multiple comorbidities. In these not treated with statins, the prognosis is highly unfavorable, even in patients without diagnosis of an active cancer.

3. Reference 12 uses a meta-analysis of many previous studies to come to a similar conclusion. The authors need to highlight what is different about their cohort or study, or what this study adds in addition to the previous studies. Do these results apply specifically to the population in this study?

Thank you for this comment. In contrast to our study, the Zhong et al. meta-analysis was performed in the general cancer population. To date, the role of statin in MI-active cancer has been studied most accurately by Yusuf et al. [24]. Our study findings have been compared with the latter reference in the Discussion section.